https://doi.org/10.1038/s41467-019-11008-z · **OPEN**

# Repeaterless quantum key distribution with efficient finite-key analysis overcoming the rate-distance limit

Kento Maeda [1,2], Toshihiko Sasaki [1] & Masato Koashi [1,2]

Quantum key distribution (QKD) over a point-to-point link enables us to benefit from a genuine quantum effect even with conventional optics tools such as lasers and photon detectors, but its capacity is limited to a linear scaling of the repeaterless bound. Recently, twin-field (TF) QKD was conjectured to beat the limit by using an untrusted central station conducting a single-photon interference detection. So far, the effort to prove the conjecture was confined to the infinite key limit which neglected the time and cost for monitoring an adversary's act. Here we propose a variant of TF-type QKD protocol equipped with a simple methodology of monitoring to reduce its cost and provide an information-theoretic security proof applicable to finite communication time. We simulate the key rate to show that the protocol beats the linear bound in a reasonable running time of sending $10^{12}$ pulses, which positively solves the conjecture.

[1] Photon Science Center, Graduate School of Engineering, The University of Tokyo, 7-3-1 Hongo, Bunkyo-ku, Tokyo 113-8656, Japan. [2] Department of Applied Physics, Graduate School of Engineering, The University of Tokyo, 7-3-1 Hongo, Bunkyo-ku, Tokyo 113-8656, Japan. Correspondence and requests for materials should be addressed to M.K. (email: koashi@qi.t.u-tokyo.ac.jp)

Quantum key distribution (QKD)[1,2] provides a secret key shared between two remote legitimate parties with information-theoretic security, enabling private communication regardless of an adversary's computational power and advanced hardware technology. It also has a welcome feature that, for a simple prepare-and-measure type of QKD protocols, the sender's and the receiver's device can be implemented with current technology such as lasers, linear optics components, and photon detectors. A drawback is a limitation on the key generation rate stemming from the loss in the channel. For a direct link from the sender to the receiver, the key rate cannot surpass the loss bounds[3,4] of $O(\eta)$, where $\eta$ is the single-photon transmissivity of the link. Although quantum repeaters[5] are known to beat this limitation by placing untrusted intermediate stations to segment the link, the required technology to manipulate quantum states is demanding. Early proposals to mitigate this demand to beat the $O(\eta)$ scaling still requires quantum memories[6] or quantum non-demolition (QND) measurements[7], which are currently in the developing stage.

Surprisingly, possibility of achieving an $O(\sqrt{\eta})$ scaling with current technology was recently proposed[8] as a protocol called twin-field (TF) QKD, a variant of the measurement-device independent (MDI) protocols[9]. In this protocol, an untrusted station Charlie sitting midway between Alice and Bob simply conducts an interference measurement to learn the relative phase between the pulse pair sent from Alice and Bob. On the surface, the scaling may be understood from the interpretation that a photon detected by Charlie has traveled either the Alice-Charlie segment or the Bob-Charlie segment with transmissivity $\sqrt{\eta}$. But a similar phase encoding scheme was already adopted in an earlier MDI-QKD protocol[10], which did not achieve the $O(\sqrt{\eta})$ scaling. The essential point lies elsewhere, in how Alice and Bob can monitor the adversary's attack on the link and on Charlie's apparatus. For this purpose, the TF QKD was specifically designed so as to attain the compatibility to the standard decoy-state method[11–13], which have been successfully used in other QKD protocols.

As the original proposal[8] lacked a rigorous security proof, many intensive studies[14–21] have been devoted to achieving information-theoretic proofs of variants of TF QKD[15–17] and a family of similar protocols called phase-matching (PM) QKD[14,18–21]. As was the case for other QKD protocols, these first proofs mainly consider the asymptotic regime. All the key rates shown to beat the loss bounds so far are achievable only in the limit of infinitely large number of pulses being sent. Explicit formulation in the finite-size regime is found only in the work of Tamaki et al.[15], but this early proposal barely surpasses the loss bounds even in the asymptotic limit, and no numerical values were given for finite-size effect. Hence, at this point, we have totally no clue on how long one must run a QKD protocol on end to beat the loss bounds. It could be hours, days, or even longer.

We should also be aware that the finite-size regime is not a mere appendage to the asymptotic regime. In the latter regime, the fraction of the communication time devoted to the monitoring of the adversary is assumed to be negligible. This implies that one is allowed to invest an infinite resource to the monitoring with no penalty, despite the fact that the monitoring is the main obstacle in the TF-type protocols. In fact, the protocol by Lin and Lütkenhaus[20], which attains both the simplest of the proofs and the highest of the asymptotic key rates, adopts a newly proposed generalization of the decoy-state method for a complete characterization of the adversary's act, by using the set of test states composed of coherent states with every complex amplitude. Although it gives a lucid view on the problem, it is probably not the shortest route to answer the ultimate question of whether one can find a protocol with information-theoretic security to beat the loss bounds with current technology.

Here we positively answer to the above question by proposing a variant of PM-QKD protocol equipped with a simple security proof in the finite-size regime. Our protocol also involves a kind of extension of the standard decoy-state method, but interestingly, its direction is the opposite of the generalization by Lin and Lütkenhaus: we try to learn about the adversary's act as little as possible except the parameter crucial for the security. For this purpose, we construct a minimal set of test states to satisfy an operator inequality, which we call an operator dominance condition. Our method drastically simplifies the analysis of the finite-size effect to just a double use of classical Bernoulli sampling.

## Results

**Proposed protocol**. The setup for our proposed protocol is illustrated in Fig. 1. In order to distribute a secure key, Alice and Bob both send optical pulses to Charlie, the central untrusted station.

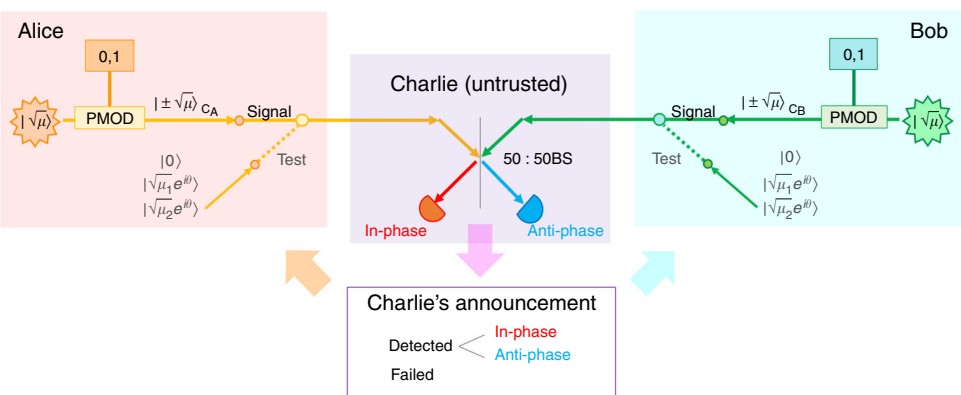

**Fig. 1** Illustration of the proposed quantum key distribution protocol. In the signal mode, Alice and Bob each encode their random bits on phase-locked pulses with intensity (mean photon number) $\mu$ through phase modulators (PMODs). In the test mode, they independently randomize the optical phase $\theta$ and switch among three intensities 0, $\mu_1$, and $\mu_2$. The central station Charlie, who may be in control of an adversary, announces whether his detection has succeeded and if it has, he further announces whether he has found the pulse pair to be in-phase or anti-phase. Bob flips his bit when anti-phase was announced

Each of the senders randomly switches between the signal mode and the test mode. They use the signal mode for accumulating raw key bits and the test mode for monitoring the amount of leak.

The signal mode is based on the PM-QKD protocol[14], which is common to previous proposals[18–21]. We assume Alice and Bob have phase-locked pulse sources to generate in-phase pulses. Each party encodes a random bit by applying 0 or $\pi$ phase shift to the pulse with a fixed intensity $\mu$ (defined in terms of its mean photon number) and sends it to Charlie. He measures and announces whether the two pulses are in-phase or anti-phase, by using a 50:50 beam-splitter and a pair of photon detectors. Successful detection at Charlie allows Alice and Bob to learn whether their bits have the same or different values. Thus, by appropriately flipping Bob's bits, Alice, and Bob can accumulate shared random bits by repetition, which we call sifted keys.

As in refs. [19,21], we associate the amount of leak to the phase errors in an equivalent protocol in which Alice and Bob use auxiliary qubits A and B. Let us call $\{|0\rangle, |1\rangle\}$ the $Z$ basis of a qubit, and $\{|\pm\rangle := (|0\rangle \pm |1\rangle)/\sqrt{2}\}$ the $X$ basis. Alice and Bob's procedure in the signal mode can be equivalently executed by preparing the qubits AB and the optical pulses $C_A C_B$ in a joint quantum state

$$\frac{|0\rangle_A |\sqrt{\mu}\rangle_{C_A} + |1\rangle_A |-\sqrt{\mu}\rangle_{C_A}}{\sqrt{2}} \otimes \frac{|0\rangle_B |\sqrt{\mu}\rangle_{C_B} - |1\rangle_B |-\sqrt{\mu}\rangle_{C_B}}{\sqrt{2}}. \quad (1)$$

Suppose that Charlie has declared $K_0$ detected rounds after the repetition. This leaves the corresponding $K_0$ pairs of qubits at Alice and Bob. If they measure the qubits in the $Z$ basis, they obtain $K_0$-bit sifted keys in the actual protocol. To assess the amount of leak in the sifted keys, we consider a virtual protocol in which they measure the qubits in the $X$ basis instead and count the number of phase errors ($X$ errors) among the $K_0$ pairs. Here an $X$ error is defined to be an event where the pair was found in either state $|+\rangle|-\rangle$ or $|-\rangle|+\rangle$. We denote the number of $X$ errors as $K_0^{(even)}$ for the reason we clarify later. If there is a promise that the phase error rate $K_0^{(even)}/K_0$ is low, it implies that the leak on the sifted keys is small. Hence, the aim of the test mode is to gather data to compute a good upper bound $e_{ph}$ on $K_0^{(even)}/K_0$. In the asymptotic limit, shortening by fraction $h(e_{ph})$ via privacy amplification achieves the security[22,23], where $h(x) = -x\log_2 x - (1-x)\log_2(1-x)$ for $x \leq 1/2$ and $h(x) = 1$ for $x > 1/2$.

To obtain a good intuition on the meaning of the observable $K_0^{(even)}$ in the virtual protocol, consider a scenario in which Alice and Bob make the $X$ basis measurements before sending out the optical pulses. Notice that the state (Eq. 1) is rewritten as

$$\left( \sqrt{c_+} |+\rangle_A |\sqrt{\mu_{even}}\rangle_{C_A} + \sqrt{c_-} |-\rangle_A |\sqrt{\mu_{odd}}\rangle_{C_A} \right)$$
$$\otimes \left( \sqrt{c_-} |+\rangle_B |\sqrt{\mu_{odd}}\rangle_{C_B} + \sqrt{c_+} |-\rangle_B |\sqrt{\mu_{even}}\rangle_{C_B} \right), \quad (2)$$

where $c_+ := e^{-\mu}\cosh\mu$ and $c_- := e^{-\mu}\sinh\mu$. The state $|\sqrt{\mu_{even}}\rangle := (|\sqrt{\mu}\rangle + |\sqrt{-\mu}\rangle)/2\sqrt{c_+}$ consists of even photon numbers, whereas the state $|\sqrt{\mu_{odd}}\rangle := (|\sqrt{\mu}\rangle - |\sqrt{-\mu}\rangle)/2\sqrt{c_-}$ consists of odd photon numbers. Then, we may interpret that an $X$ error occurs with probability $p_{even} := c_+^2 + c_-^2 = e^{-2\mu}\cosh 2\mu$ and the optical pulses are sent in state $\rho^{(even)}$, which is given by

$$p_{even}\rho^{(even)} = c_+^2 |\sqrt{\mu_{even}}\sqrt{\mu_{even}}\rangle\langle\sqrt{\mu_{even}}\sqrt{\mu_{even}}|$$
$$+ c_-^2 |\sqrt{\mu_{odd}}\sqrt{\mu_{odd}}\rangle\langle\sqrt{\mu_{odd}}\sqrt{\mu_{odd}}|. \quad (3)$$

For probability $p_{odd} := 1 - p_{even}$, the optical pulses are sent in state

$\rho^{(odd)}$, where

$$p_{odd}\rho^{(odd)} = c_+ c_- |\sqrt{\mu_{even}}\sqrt{\mu_{odd}}\rangle\langle\sqrt{\mu_{even}}\sqrt{\mu_{odd}}|$$
$$+ c_- c_+ |\sqrt{\mu_{odd}}\sqrt{\mu_{even}}\rangle\langle\sqrt{\mu_{odd}}\sqrt{\mu_{even}}|. \quad (4)$$

We see that for state $\rho^{(even)}$, the total number of photons in the pulse pair is always even. Hence, the number $K_0^{(even)}$ can be interpreted as the frequency of detection when the total emitted photon number of the pulse pair was even.

The main question is how we should design the test mode to estimate the number $K_0^{(even)}$ in the signal mode. An obvious choice is to prepare actually the state $\rho^{(even)}$ as was proposed recently[21], but generation of such a non-classical optical state with a good fidelity will be hard to realize in current technology. For the use of laser pulses, previous approaches[18,19] for the asymptotic regime use the standard decoy-state method in which various detection rates labeled by emitted photon numbers are estimated. A bound on the phase error rate is then computed from those rates through a set of inequalities. Lin and Lütkenhaus[20] generalized the decoy-state method to a kind of tomography, in which case tight estimation of phase error rate $K_0^{(even)}/K_0$ should be possible. In order to simplify the security argument for the finite-size regime, here we take a more direct approach of constructing a state approximating $\rho^{(even)}$. Of course, $\rho^{(even)}$ is a highly non-classical optical state and thus it is impossible to approximate it by a mixture of coherent states. As the second-best plan, we propose to find a linear combination $\sum_i \alpha^{(i)}\rho^{(i)}(\alpha^{(i)} \in \mathbb{R})$ of test states $\{\rho^{(i)}\}$ to approximate $\rho^{(even)}$. The crux is that we allow coefficients $\{\alpha^{(i)}\}$ to include negative values as long as it satisfies an operator inequality,

$$\sum_i \alpha^{(i)}\rho^{(i)} \geq \rho^{(even)}, \quad (5)$$

which we call an operator dominance condition.

Based on the above design policy, we found the following potocol (see Fig. 1).

1. Alice chooses a label from {"0", "10", "11", "2"} with probabilities $p_0$, $p_{10}$, $p_{11}$, and $p_2$, respectively. According to the label, Alice performs one of the following procedures.
   "0": She generates a random bit $a$ and sends a pulse with amplitude $(-1)^a\sqrt{\mu}$.
   "10": She sends the vacuum.
   "11": She sends a phase-randomized pulse with intensity $\mu_1$.
   "2": She sends a phase-randomized pulse with intensity $\mu_2$.
2. Bob independently carries out the same procedure as Alice in Step 1.
3. Alice and Bob repeat Steps 1 and 2 in total of $N_{tot}$ times.
4. For every pair of pulses received from Alice and Bob, Charlie announces whether the phase difference was successfully detected. When it was detected, he further announces whether it was in-phase or anti-phase.
5. Alice and Bob disclose their label choices. Let $K_0$ be the number of detected rounds for which both Alice and Bob chose "0". Alice concatenates the random bits for the $K_0$ rounds to define her sifted key. Bob defines his sifted key in the same way except that he flips all the bits for the rounds declared to be anti-phase.
6. Let $K_{10}$, $K_{11}$, and $K_2$ be the number of detected rounds for which both Alice and Bob chose the same label "10", "11", and "2", respectively. Let $K_1 := K_{10} + K_{11}$.
7. For error correction, Alice announces $H_{EC}$ bits of syndrome of a linear code for her sifted key. Bob reconciles his sifted

key accordingly. Alice and Bob verify the correction by comparing $\zeta'$ bits via universal$_2$ hashing[24].

8. They apply the privacy amplification to obtain final keys of length

$$G = K_0 - \lceil K_0 h(f(K_1, K_2)/K_0) \rceil - H_{EC} - \zeta - \zeta', \quad (6)$$

where the parameter $\zeta$ and the function $f(K_1, K_2)$ will be specified below.

**Security proof**. In order to prove the security of the above protocol, we need to construct an upper bound on the phase error rate $K_0^{(even)}/K_0$ in the virtual protocol. To cover the finite-size cases as well, our objective is to construct $f(K_1, K_2)$ which satisfies

$$\text{Prob}\left\{ K_0^{(even)} \leq f(K_1, K_2) \right\} \geq 1 - \epsilon \quad (7)$$

for any attack in the virtual protocol. It is known that it immediately implies that the actual protocol is $\epsilon_{sec}$-secure with a small security parameter $\epsilon_{sec} = \sqrt{2}\sqrt{\epsilon + 2^{-\zeta}} + 2^{-\zeta'}$. See methods section for the detailed definition of security.

Let $\tau(\mu)$ be the phase-randomized coherent state with mean photon number $\mu$,

$$\tau(\mu) := \int_0^{2\pi} d\theta \left| \sqrt{\mu} e^{i\theta} \right\rangle \left\langle \sqrt{\mu} e^{i\theta} \right| = \sum_{n=0}^{\infty} \frac{\mu^n e^{-\mu}}{n!} |n\rangle\langle n|. \quad (8)$$

Our proof method is based on an operator dominance condition which reads

$$p_{10}^2 \tau(0) \otimes \tau(0) + p_{11}^2 \tau(\mu_1) \otimes \tau(\mu_1) - \Gamma \tau(\mu_2) \otimes \tau(\mu_2) \geq \Lambda \rho^{(even)}, \quad (9)$$

where $\Gamma$ and $\Lambda$ are positive constants. Our security argument below holds for any set of parameters $(p_{10}, p_{11}, \mu, \mu_1, \mu_2, \Gamma, \Lambda)$ satisfying Eq. (9). A simple method of computing $\Gamma$ and $\Lambda$ from $(p_{10}, p_{11}, \mu, \mu_1, \mu_2)$ is given in methods section.

We first clarify the meaning of numbers $K_1$ and $K_2$ collected in the test mode. By definition of the protocol, $K_1$ is the frequency of detection when the pulse pair $C_A C_B$ was initially prepared in state $\rho^{(test1)}$, where

$$(p_{10}^2 + p_{11}^2)\rho^{(test1)} = p_{10}^2 \tau(0) \otimes \tau(0) + p_{11}^2 \tau(\mu_1) \otimes \tau(\mu_1). \quad (10)$$

Similarly, $K_2$ is the frequency of detection for state

$$\rho^{(test2)} = \tau(\mu_2) \otimes \tau(\mu_2). \quad (11)$$

Also recall that $K_0^{(even)}$ is the frequency of detection for state $\rho^{(even)}$ defined in Eq. (3).

When Eq. (9) holds, there exists a normalized state $\rho^{(junk)}$, which satisfies

$$(p_{10}^2 + p_{11}^2)\rho^{(test1)} = \Gamma\rho^{(test2)} + \Lambda\rho^{(even)} + \Delta\rho^{(junk)} \quad (12)$$

for $\Delta := p_{10}^2 + p_{11}^2 - \Gamma - \Lambda \geq 0$. Therefore, we can reinterpret the state $\rho^{(test1)}$ as a mixture of the three states $\rho^{(test2)}$, $\rho^{(junk)}$, and $\rho^{(even)}$. Let us consider a modified scenario in which the state of the pulse pair is directly prepared in various states with the probabilities specified in Fig. 2. In this scenario, the frequencies $K_1^{(test2)}$ and $K_1^{(even)}$ shown in Fig. 2 are also well-defined. Suppose that the adversary's attack (which may include taking over Charlie's announcement) is the same as that for the actual/virtual protocols. As the breakdown of the mixed state $\rho^{(test1)}$ in the actual protocol is revealed only after Charlie has announced all the detections, we see that the following property naturally holds.

(i)  The marginal joint probability of the three variables $(K_2, K_1, K_0^{(even)})$ in the modified scenario is the same as that in the virtual protocol.

This means that if Eq. (7) is true in the modified scenario, it is also true in the virtual protocol.

From comparison between the first and the second rows in Fig. 2, we notice that $K_2$ and $K_1^{(test2)}$ in the modified scenario are detection frequencies of the same initial state $\rho^{(test2)}$. As the adversary has no clue about whether a pulse pair in state $\rho^{(test2)}$ belongs to Test1 mode or to Test2 mode, they cannot force Charlie to detect one of the cases preferably over the others. Hence, the ratio of $K_2$ to $K_1^{(test2)}$ is expected to be close to the initial ratio of the two cases, $p_2^2/\Gamma$. More precisely, $K_2$ is a Bernoulli sampling from a population with $K_2 + K_1^{(test2)}$ elements. This is also the case with $K_1^{(even)}$ and $K_0^{(even)}$. It leads to the following property of conditional probabilities stated in terms of binomial distribution $B(K; n, p) := p^K (1-p)^{n-K} n!/K!(n-K)!$.

(ii)  In the modified scenario, it holds that
$$\text{Prob}\left\{ K_2 | K_2 + K_1^{(test2)} = n \right\} = B(K_2; n, p_2^2/(p_2^2 + \Gamma)) \quad (13)$$

and similarly,

$$\text{Prob}\left\{ K_1^{(even)} | K_0^{(even)} + K_1^{(even)} = n \right\} = B\left( K_1^{(even)}; n, \Lambda/(p_0^2 p_{even} + \Lambda) \right). \quad (14)$$

The properties (i) and (ii) reduce the security proof to an elementary problem of classical random sampling. In an asymptotic limit of $K_1, K_2 \to \infty$, a bound on $K_0^{(even)}$ is immediately obtained from the relations $K_1^{(test2)}/K_2 = \Gamma/p_2^2$, $K_0^{(even)}/K_1^{(even)} = p_0^2 p_{even}/\Lambda$, and $K_1 \geq K_1^{(test2)} + K_1^{(even)}$. A finite-size bound $f(K_1, K_2)$ satisfying Eq. (7) can be constructed by the use of the Chernoff bound[25]. As explained in methods section, we can compute general bounds $M^\pm(K, p, \epsilon)$ that satisfy

$$\text{Prob}\{M \leq M^+(K; p, \epsilon)\} \geq 1 - \epsilon \quad (15)$$

$$\text{Prob}\{M \geq M^-(K; p, \epsilon)\} \geq 1 - \epsilon, \quad (16)$$

when $\text{Prob}\{K | M + K = n\} = B(K; n, p)$ holds for all $n \geq 1$. Then, we can construct the function $f(K_1, K_2)$ as

$$f(K_1, K_2) = M^+\left( K_1^{(even)+}; \frac{\Lambda}{p_0^2 p_{even} + \Lambda}, \frac{\epsilon}{2} \right) \quad (17)$$

with

$$K_1^{(even)+} := K_1 - M^-\left( K_2; \frac{p_2^2}{p_2^2 + \Gamma}, \frac{\epsilon}{2} \right) \quad (18)$$

which obviously satisfies Eq. (7) and hence completes the security proof.

For an intuitive understanding of the amount of the finite-size effect, an approximate expression of the bound $f(K_1, K_2)$ may be helpful. The general bounds $M^\pm$ are approximated as

$$M^\pm(K; p, \epsilon) \cong \frac{1-p}{p} K \pm \sqrt{-\log\epsilon} \frac{\sqrt{2(1-p)}}{p} \sqrt{K} \quad (19)$$

when $(1-p)K \gg -\log\epsilon$. Then, we can approximate $f(K_1, K_2)$ as

$$f(K_1, K_2) = \frac{p_0^2 p_{even}}{\Lambda} \left( K_1 - \frac{\Gamma}{p_2^2} K_2 + \nu(K_1, K_2)\sqrt{-\log(\epsilon/2)} \right) \quad (20)$$

| Actual/virtual protocol | | | | Mode | Modified scenario | | | | |
|---|---|---|---|---|---|---|---|---|---|
| Local label | Detection frequency | Initial probability | Quantum state | | Quantum state | Initial probability | Detection frequency | | |
| "2","2" | $K_2$ | $p_2 \times p_2$ | $\tau(\mu_2) \otimes \tau(\mu_2)$ | Test2 | $\rho^{(\text{test2})}$ | $p_2^2$ | $K_2$ | | Binomial distribution |
| "10","10" | $K_1$ | $p_{10} \times p_{10}$ | $\tau(0) \otimes \tau(0)$ | Test1 | $\rho^{(\text{test2})}$ | $\Gamma$ | $K_1^{(\text{test2})}$ | $K_1$ | |
| | | | | | $\rho^{(\text{junk})}$ | $\Delta$ | | | |
| "11","11" | | $p_{11} \times p_{11}$ | $\tau(\mu_1) \otimes \tau(\mu_1)$ | | $\rho^{(\text{even})}$ | $\Lambda$ | $K_1^{(\text{even})}$ | | Binomial distribution |
| "0","0" | $K_0$ | $K_0^{(\text{even})}$ $p_0^2 p_{\text{even}}$ | $\rho^{(\text{even})}$ | Signal | $\rho^{(\text{even})}$ | $p_0^2 p_{\text{even}}$ | $K_0^{(\text{even})}$ | $K_0$ | Binomial distribution |
| | | $K_0^{(\text{odd})}$ $p_0^2 p_{\text{odd}}$ | $\rho^{(\text{odd})}$ | | $\rho^{(\text{odd})}$ | $p_0^2 p_{\text{odd}}$ | $K_0^{(\text{odd})}$ | | |
| ⋮ | ⋮ | ⋮ | ⋮ | ⋮ | ⋮ | ⋮ | ⋮ | | |

**Fig. 2** Relation between the actual/virtual protocol and the modified scenario. Each row is chosen with the initial probability and the pulse pair is prepared in the corresponding quantum state. The detection frequency is the number of times Charlie has declared success. The cases when Alice's and Bob's label differ are irrelevant and not shown. In the actual protocol, three detection frequencies, $K_0$, $K_1$, and $K_2$ are determined. In the virtual protocol, $K_0$ is decomposed into a sum of two frequencies, $K_0^{(\text{even})}$ and $K_0^{(\text{odd})}$. The security of the actual protocol is quantitatively assured if a good upper bound on $K_0^{(\text{even})}$ is found. To find such a bound, we consider a modified scenario in which the variables $(K_2, K_1, K_0^{(\text{even})})$ follows the same statistics as in the virtual protocol. In the modified scenario, $K_1$ is interpreted as a sum of three frequencies corresponding to three different initial quantum states of the pulse pair, $\rho^{(\text{test2})}$, $\rho^{(\text{junk})}$, and $\rho^{(\text{even})}$. We notice that the first two rows are chosen with probabilities $p_2^2$ and $\Gamma$ and classified to the Test2 mode and to the Test1 mode accordingly, but the pulse pairs are initially prepared in the same state $\rho^{(\text{test2})}$. Charlie's success/failure declaration and the Test2/Test1 mode choice should thus be statistically independent. It follows that the conditional statistics of variable $K_2$ obeys a Binomial distribution given that the sum $K_2 + K_1^{(\text{test2})}$ is a constant. This leads to a lower bound on $K_1^{(\text{test2})}$ in terms of $K_2$. A similar argument holds for variables $K_1^{(\text{even})}$ and $K_0^{(\text{even})}$, leading to an upper bound on $K_0^{(\text{even})}$ in terms of $K_1^{(\text{even})}$. Combining these, we obtain an upper bound on $K_0^{(\text{even})}$ in terms of $K_1$ and $K_2$, which should be applicable to the virtual protocol

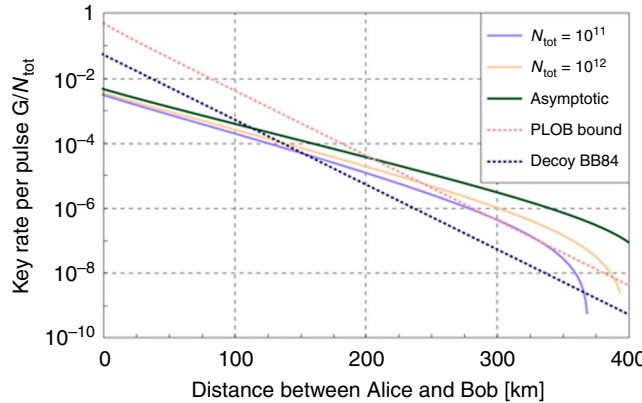

**Fig. 3** The key rate per pulse as a function of distance $L$ between Alice and Bob. We assumed a fiber loss of 0.2 dB/km, a loss-independent misalignment error of $e_m = 0.03$, a detector dark counting probability of $p_d = 10^{-8}$, and a detection efficiency of $\eta_d = 0.3$. The rate in the asymptotic limit and those in the finite-size cases with transmission of $N_{\text{tot}} = 10^{11}$, $10^{12}$ pulse pairs are shown. For comparison, we also show the PLOB bound[4] and the asymptotic key rate of ideal decoy-BB84 protocol[2,4] for the direct link transmissivity $\eta_d 10^{-0.2L/10}$

with

$$v(K_1, K_2) \cong \left[ \frac{\sqrt{2\Gamma(p_2^2 + \Gamma)}}{p_2^2} \sqrt{K_2} \right. $$
$$\left. + \sqrt{2\left(1 + \frac{\Lambda}{p_0^2 p_{\text{even}}}\right)} \sqrt{K_1 - \frac{\Gamma}{p_2^2} K_2} \right]. \qquad (21)$$

**Numerical simulation.** We simulated the key rate $G/N_{\text{tot}}$ as a function of distance $L$ between Alice and Bob when they are fiber-linked to Charlie with a loss of 0.2 dB/km. We assumed a detection efficiency of $\eta_d = 0.3$ for Charlie's apparatus. The parameters $(\mu, \mu_1, \mu_2, p_0, p_{10}, p_{11}, p_2)$ are optimized for each

distance. The detail of the model for determining $K_0$, $K_1$, and $K_2$ is given in methods section.

Figure 3 shows the key rates of our protocol in the asymptotic limit and in the finite-size cases with $N_{\text{tot}} = 10^{11}$ and $10^{12}$. We have also plotted the PLOB bound[4], $-\log_2(1 - \eta_{\text{AB}})$, for the direct link from Alice to Bob with transmissivity $\eta_{\text{AB}} = \eta_d 10^{-0.2L/10}$, assuming the same detection efficiency. The asymptotic key rate shows an $O(\sqrt{\eta_{\text{AB}}})$ scaling. As expected, the asymptotic rate is lower than those of the protocols[18–20] investing more resources for the monitoring. The main feature of our protocol lies in the provably secure key rate in the finite-size regime. We see that at $N_{\text{tot}} = 10^{11}$ it barely surpasses the PLOB bound, and at $N_{\text{tot}} = 10^{12}$ it clearly beats the bound at ~ 300 km. The dotted line below the PLOB bound is the asymptotic rate for the ideal decoy-state BB84 protocol[2,4,13], $\eta_{\text{AB}}/(2e)$, which is surpassed by our protocol beyond 200 km even with $N_{\text{tot}} = 10^{11}$.

As an example, we present explict values of the optimized parameters for $N_{\text{tot}} = 10^{12}$ at 340 km. The intensities are $(\mu, \mu_1, \mu_2) = (0.012, 0.23, 0.022)$ and the probabilities are $(p_0, p_{10}, p_{11}, p_2) = (0.73, 0.21, 0.013, 0.049)$. The operator dominance condition (Eq. 9) is satisfied with $\Gamma = 1.2 \times 10^{-3}$ and $\Lambda = 1.2 \times 10^{-2}$. The observed values expected from the model are $(K_0, K_1, K_2) = (1.6 \times 10^6, 1.0 \times 10^5, 1.2 \times 10^5)$.

## Discussion
We proposed a variation of TF-type QKD protocol by using the signal mode of the PM-QKD protocol and the test mode specifically designed to simplify the estimation process of the amount of information leak. The simulated key rate shows that it beats the PLOB bound when the total number of pulse pairs emitted from Alice and Bob is $10^{11}$ to $10^{12}$, which corresponds to several to twenty minutes for a system of 1 GHz pulse repetition. It amounts to settling down the conjecture with a comprehensive information-theoretic security proof covering the finite-size key regime.

In the protocol, the events where Alice and Bob have chosen different local labels are simply discarded. It is an interesting

question whether we may improve the key rate by incorporating the detection frequencies of such events in the analysis. Conversely, by accepting a lower key rate, we may be able to simplify the protocol to use only three intensities $(0, \mu, \mu_1)$ instead of four in the current protocol. We leave these questions to future study.

An essential ingredient of our design is the operator dominance method of estimating the detection frequency of one state from those of a combination of different test states. We can identify two instances of binomial distribution in a modified scenario, which simplifies the required statistical analysis in the finite-size regime. As a methodology, the number of test states forming the linear combination to approximate the target state does not affect the simplicity of analysis. As long as the operator dominance condition is satisfied, we can group the states with positive coefficients to define state $\rho^{(\text{test1})}$ and those with negative to define $\rho^{(\text{test2})}$. Such a flexibility will be used to improve the finite-size key rate of TF-type protocols further. We also expect that the method can be used to simplify the security analysis of other QKD protocols, especially when the imperfection of practical devices is taken into account.

## Methods

**Definition of security in the finite-size regime.** We evaluate the secrecy of the final key as follows. When the final key length is $G \geq 1$, we represent Alice's final key and an adversary's quantum system as a joint state

$$\rho_{\text{AE}|G}^{\text{fin}} = \sum_{z=0}^{2^G-1} \text{Prob}(z)|z\rangle\langle z|_A \otimes \rho_{\text{E}|G}^{\text{fin}}(z), \tag{22}$$

and define the corresponding ideal state as

$$\rho_{\text{AE}|G}^{\text{ideal}} = \sum_{z=0}^{2^G-1} 2^{-G}|z\rangle\langle z|_A \otimes \text{Tr}_A\left(\rho_{\text{AE}|G}^{\text{fin}}\right). \tag{23}$$

Let $\|\sigma\|_1 = \text{Tr}\sqrt{\sigma^\dagger \sigma}$ be the trace norm of an operator $\sigma$. We say a protocol is $\epsilon_{\text{sct}}$-secret when

$$\frac{1}{2}\sum_{G\geq 1}\text{Prob}(G)\parallel \rho_{\text{AE}|G}^{\text{fin}} - \rho_{\text{AE}|G}^{\text{ideal}} \parallel_1 \leq \epsilon_{\text{sct}} \tag{24}$$

holds regardless of the adversary's attack. It is known[26] that if the number of phase errors is bounded as in Eq. (7), the protocol is $\epsilon_{\text{sct}}$-secret with $\epsilon_{\text{sct}} = \sqrt{2}\sqrt{\epsilon + 2^{-\zeta}}$.

For correctness, we say a protocol is $\epsilon_{\text{cor}}$-correct if the probability for Alice's and Bob's final key to differ is bounded by $\epsilon_{\text{cor}}$. Our protocol achieves $\epsilon_{\text{cor}} = 2^{-\zeta'}$ via the verification in Step 7.

When the above two conditions are met, the protocol becomes $\epsilon_{\text{sec}}$-secure with $\epsilon_{\text{sec}} = \epsilon_{\text{sct}} + \epsilon_{\text{cor}}$ in the sense of universal composability[27].

**Construction of operator dominance condition.** Here we describe a procedure to compute parameter sets fulfilling the operator dominance condition (Eq. 9). Suppose that values of $\mu_1, \mu_2, p_{10}$, and $p_{11} > 0$ satisfying

$$0 < \frac{\mu_1 - \mu_2}{\mu_2} < \frac{p_{10}^2}{p_{11}^2 e^{-2\mu_1}} \tag{25}$$

are given. Then, we can satisfy Eq. (9) by choosing $\Gamma$ and $\Lambda$ according to the following:

$$\frac{\Gamma}{p_{11}^2} = \frac{\mu_1 e^{-2\mu_1}}{\mu_2 e^{-2\mu_2}} \tag{26}$$

$$\frac{p_{\text{even}}p_{11}^2}{\Lambda} = \frac{e^{-2\mu}}{p_{10}^2/p_{11}^2 - e^{-2\mu_1}(\mu_1 - \mu_2)/\mu_2} + \frac{e^{-2\mu}}{\mu_1 e^{-2\mu_1}}\sum_{k=1}^{\infty}\frac{(k+1)\mu^{2k}}{\mu_1^{2k-1} - \mu_2^{2k-1}}. \tag{27}$$

The proof goes as follows. Using the representation $\tau(\mu) = e^{-\mu}\sum_k (\mu^k/k!)|k\rangle\langle k|$, we see that the lefthand side of Eq. (9) has a diagonal form $\sum_{k,k'}(q_{k+k'}/k!k'!)|k,k'\rangle\langle k,k'|$ on the Fock basis, where

$$q_m = \begin{cases} p_{11}^2 e^{-2\mu_1}\mu_1^m - \Gamma e^{-2\mu_2}\mu_2^m & (m \geq 1) \\ p_{11}^2 e^{-2\mu_1} - \Gamma e^{-2\mu_2} + p_{10}^2 & (m = 0). \end{cases} \tag{28}$$

Substituting Eq. (26), we have

$$q_m = \begin{cases} p_{11}^2\mu_1 e^{-2\mu_1}\left(\mu_1^{m-1} - \mu_2^{m-1}\right) > 0 & (m \geq 2) \\ 0 & (m = 1) \\ p_{10}^2 - p_{11}^2 e^{-2\mu_1}(\mu_1 - \mu_2)/\mu_2 > 0 & (m = 0) \end{cases} \tag{29}$$

under condition (Eq. 25). Using $q_m$, Eq. (27) is rewritten as

$$\frac{p_{\text{even}}}{\Lambda} = e^{-2\mu}\sum_{k=0}^{\infty}\frac{(k+1)\mu^{2k}}{q_{2k}}. \tag{30}$$

Let $\pi_{\text{e}} = \sum_{k=0}^{\infty}|2k\rangle\langle 2k|$ and $\pi_{\text{o}} = \sum_{k=0}^{\infty}|2k+1\rangle\langle 2k+1|$ be projections to the subspaces with even and odd photon numbers, respectively. We denote $\pi_{st} := \pi_s \otimes \pi_t (s,t = \text{e}, \text{o})$. From Eq. (3), we have

$$p_{\text{even}}\rho^{(\text{even})} = \pi_{\text{ee}}|\sqrt{\mu}, \sqrt{\mu}\rangle\langle\sqrt{\mu}, \sqrt{\mu}|\pi_{\text{ee}} + \pi_{\text{oo}}|\sqrt{\mu}, \sqrt{\mu}\rangle\langle\sqrt{\mu}, \sqrt{\mu}|\pi_{\text{oo}}. \tag{31}$$

Hence, Eq. (9) is equivalent to the following set of conditions:

$$p_{\text{even}}\sum_{k,k':\text{even}}\frac{q_{k+k'}}{k!k'!}|k,k'\rangle\langle k,k'| \geq \Lambda\pi_{\text{ee}}|\sqrt{\mu}, \sqrt{\mu}\rangle\langle\sqrt{\mu}, \sqrt{\mu}|\pi_{\text{ee}} \tag{32}$$

$$p_{\text{even}}\sum_{k,k':\text{odd}}\frac{q_{k+k'}}{k!k'!}|k,k'\rangle\langle k,k'| \geq \Lambda\pi_{\text{oo}}|\sqrt{\mu}, \sqrt{\mu}\rangle\langle\sqrt{\mu}, \sqrt{\mu}|\pi_{\text{oo}} \tag{33}$$

$$q_{k+k'} \geq 0 \ (k+k' : \text{odd}). \tag{34}$$

The condition (Eq. 34) is obviously true from Eq. (29). Since $q_{k+k'} > 0$ when $k + k'$ is even, Eq. (32) is true if

$$p_{\text{even}}\pi_{\text{ee}} \geq \Lambda|\varphi_{\text{ee}}\rangle\langle\varphi_{\text{ee}}| \tag{35}$$

with

$$|\varphi_{\text{ee}}\rangle = \sum_{k,k':\text{even}}\left(\frac{q_{k+k'}}{k!k'!}\right)^{-1/2}|k,k'\rangle\langle k,k'|\sqrt{\mu}, \sqrt{\mu}\rangle. \tag{36}$$

Since

$$\langle\varphi_{\text{ee}}|\varphi_{\text{ee}}\rangle = \sum_{k,k':\text{even}}\frac{e^{-2\mu}\mu^{k+k'}}{q_{k+k'}} = \frac{p_{\text{even}}}{\Lambda} \tag{37}$$

from Eq. (30), we see that condition Eq. (35) is true and so is condition (Eq. 32). Similarly, for

$$|\varphi_{\text{oo}}\rangle = \sum_{k,k':\text{odd}}\left(\frac{q_{k+k'}}{k!k'!}\right)^{-1/2}|k,k'\rangle\langle k,k'|\sqrt{\mu}, \sqrt{\mu}\rangle, \tag{38}$$

we have

$$\langle\varphi_{\text{oo}}|\varphi_{\text{oo}}\rangle = e^{-2\mu}\sum_{k=1}^{\infty}\frac{k\mu^{2k}}{q_{2k}} < \frac{p_{\text{even}}}{\Lambda}, \tag{39}$$

implying that condition (Eq. 33) is also true.

**Bounds for a classical random sampling.** Here we give a computable definition of functions $M^\pm(K; p, \epsilon)$ and prove the relevant properties. We assume $p \in (0, 1)$ and $\epsilon > 0$. Let $\bar{p} := 1 - p$, $M_{p,K} := K\bar{p}/p$, $K_{p,\epsilon} := \log\epsilon/\log p$, and

$$g(M, K) := (M + K)D(M/(M + K) \parallel \bar{p}) \tag{40}$$

with $D(q \parallel p) := q\log(q/p) + (1 - q)\log[(1 - q)/(1 - p)]$. Then, for $K \geq 0$, we have $g(0, K_{p,\epsilon}) = -\log\epsilon$, $g(M_{p,K}, K) = 0$, and $g(\infty, K) = \infty$. The partial derivatives satisfy

$$\frac{\partial g}{\partial M} > 0, \frac{\partial g}{\partial K} < 0 \quad \text{for} \quad M > (M + K)\bar{p} \tag{41}$$

and

$$\frac{\partial g}{\partial M} < 0, \frac{\partial g}{\partial K} > 0 \quad \text{for} \quad M < (M + K)\bar{p}. \tag{42}$$

Hence we may uniquely define $M^\pm(K; p, \epsilon)$ for $K \geq 0$ as follows.
*Definition 1*
$M^+$ is the unique solution of the equation $g(M, K) = -\log\epsilon$ for $M \in (M_{p,K}, \infty)$. For $K > K_{p,\epsilon}$, $M^-$ is the unique solution of the equation $g(M, K) = -\log\epsilon$ for $M \in (0, M_{p,K})$. For $K \leq K_{p,\epsilon}$, let $M^- := 0$.

Due to the properties of $g(M, K)$ described above, $M^\pm(K)$ is non-decreasing. Using this definition, we can prove the following lemma:
*Lemma 1*
Let $M$ and $K$ be random variables taking nonnegative integer values. If $\text{Prob}\{K|M + K = n\} = B(K; n, p)$ for all $n \geq 1$, then

$$\text{Prob}\{M \leq M^+(K; p, \epsilon)\} \geq 1 - \epsilon \tag{43}$$

and

$$\mathrm{Prob}\{M \geq M^{-}(K; p, \epsilon)\} \geq 1 - \epsilon. \tag{44}$$

Proof: using the Chernoff bound[25] for the binominal distribution, we have

$$\mathrm{Prob}\{nD(M/n \parallel \bar{p}) \geq -\log\epsilon \;\wedge\; M \geq n\bar{p}|M + K = n\} \leq \epsilon \tag{45}$$

for all $n \geq 1$, leading to

$$\mathrm{Prob}\{g(M, K) \geq -\log\epsilon \;\wedge\; M \geq (M + K)\bar{p} \;\wedge\; M + K \neq 0\} \leq \epsilon. \tag{46}$$

If $M > M^{+}(K) > M_{p,K}$, then $M > (M + K)\bar{p} \geq 0$ and $g(M, K) > g(M^{+}(K), K) = -\log\epsilon$ hold. Hence Eq. (46) implies $\mathrm{Prob}\{M > M^{+}(K)\} \leq \epsilon$, leading to Eq. (43). Similarly to Eq. (46), we can also obtain

$$\mathrm{Prob}\{g(M, K) \geq -\log\epsilon \;\wedge\; M \leq (M + K)\bar{p} \;\wedge\; M + K \neq 0\} \leq \epsilon. \tag{47}$$

If $M < M^{-}(K) < M_{p,K}$, then $M < (M + K)\bar{p}$, $K > K_{p,\epsilon} > 0$, and $g(M, K) > g(M^{+}(K), K) = -\log\epsilon$ hold. Then, Eq. (47) implies $\mathrm{Prob}\{M < M^{-}(K)\} \leq \epsilon$, leading to Eq. (44).

**Calculation of simulated key rates**. For the simulation of the key rate $G/N_{\mathrm{tot}}$ as a function of distance between Alice and Bob, we adopted the following model for the channels and Charlie's detection apparatus. We assumed a fiber loss of 0.2 dB/km and a detection efficiency of $\eta_{\mathrm{d}} = 0.3$ for Charlie's apparatus. The distance between Alice and Bob is denoted by $L$ (in km). The overall transmissivity from Alice to Charlie's detection is then $\eta = \eta_{\mathrm{d}}10^{-0.2L/20}$. The overall transmissivity from Bob to Charlie is also $\eta$. We assume that (honest) Charlie declares a success when one or both of the detectors have reported detection. When both have detected, he randomly declares in-phase or anti-phase. We assume that each detector has a dark count probability of $p_{\mathrm{d}} = 10^{-8}$, which amounts to the effective probability $d := 2p_{\mathrm{d}} - p_{\mathrm{d}}^2$ from the two detectors. The expected frequencies of detection are then modeled as

$$K_0/N_{\mathrm{tot}} = p_0^2(1 - e^{-2\eta\mu} + e^{-2\eta\mu}d), \tag{48}$$

$$K_1/N_{\mathrm{tot}} = p_{11}^2(1 - e^{-2\eta\mu_1} + e^{-2\eta\mu_1}d) + p_{10}^2 d, \tag{49}$$

$$K_2/N_{\mathrm{tot}} = p_2^2(1 - e^{-2\eta\mu_2} + e^{-2\eta\mu_2}d). \tag{50}$$

For the bit error rate, we use the following model that includes a mode/phase mismatch error of $e_{\mathrm{m}} = 0.03$:

$$e_{\mathrm{bit}} = \frac{\left[1 - \sqrt{1 - d}\exp(-2e_{\mathrm{m}}\eta\mu)\right]\left[1 + \sqrt{1 - d}\exp(-2(1 - e_{\mathrm{m}})\eta\mu)\right]}{2[1 - (1 - d)e^{-2\eta\mu}]}. \tag{51}$$

We assume the cost of error correction $H_{\mathrm{EC}}$ to be $1.1 \times K_0 h(e_{\mathrm{bit}})$.

For the calculation of the key rate with a finite value of $N_{\mathrm{tot}}$, we chose the security parameters as $\epsilon = 2^{-66}$, $\zeta = 66$, and $\zeta' = 32$, which makes the protocol $\epsilon_{\mathrm{sec}}$-secure with $\epsilon_{\mathrm{sec}} = 2^{-31} < 10^{-10}$. The final key length $G = K_0(1 - h(f(K_1, K_2)/K_0)) - H_{\mathrm{EC}} - \zeta - \zeta'$ is then optimized with the Nelder–Mead method over six parameters $\mu, a = \mu_1/\mu, b = \mu_2/\mu, p_2, p_1 = p_{10} + p_{11}$, and $s = p_{10}/(p_{10} + p_{11})$. For every point shown in Fig. 3, we confirmed that the absolute values of the numerical partial derivative at each optimized condition were sufficiently small compared with the parameter values.

For calculation of the asymptotic key rate, we analytically reduced the number of parameters as follows. Using Eq. (20), the phase error rate for $N_{\mathrm{tot}}, K_0, K_1, K_2 \to \infty$ is given by

$$\frac{f(K_1, K_2)}{K_0} = \frac{p_0^2}{K_0}\frac{p_{\mathrm{even}}p_{11}^2}{\Lambda}\left(\frac{K_1}{p_{11}^2} - \frac{\Gamma}{p_{11}^2}\frac{K_2}{p_2^2}\right). \tag{52}$$

From Eqs. (26), (27), (48), (49), and (50), we see that it can be cast into the form $f(K_1, K_2)/K_0 = g(p_{10}^2/p_{11}^2)$ with

$$g(\lambda) = C_1\left(\frac{1}{\lambda - C_2} + C_3\right)(\lambda + C_4), \tag{53}$$

where $\{C_j\}_j$ depend only on $\mu, \mu_1, \mu_2, \eta$, and $d$. The function $g(\lambda)$ takes its minimum at $\lambda^* := C_2 + \sqrt{(C_2 + C_4)/C_3}$ with

$$g(\lambda^*) = C_1\left(1 + \sqrt{(C_2 + C_4)C_3}\right)^2. \tag{54}$$

Hence, in the limit of $p_0 \to 1$ and $p_{10}, p_{11}, p_2 \to 0$ with $p_{10}^2/p_{11}^2 = \lambda^*$, we have

$$\frac{G}{N_{\mathrm{tot}}} \to \left(1 - e^{-2\eta\mu} + e^{-2\eta\mu}d\right)(1 - h(g(\lambda^*))) - 1.1 \times h(e_{\mathrm{bit}}). \tag{55}$$

To calculate the asymptotic key rate in Fig. 3, we optimized the above expression over $\mu, a = \mu_1/\mu$ and $b = \mu_2/\mu$ with the Nelder–Mead method.

## Data availability
Data sharing not applicable to the article as no data sets were generated or analyzed during the current study.

## Code availability
Computer codes to calculate the key rates are available from the corresponding author upon reasonable request.

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

## Acknowledgements
We thank Kiyoshi Tamaki and Koji Azuma for valuable discussions. This work was supported by Cross-ministerial Strategic Innovation Promotion Program (SIP) (Council

for Science, Technology and Innovation (CSTI)); ImPACT Program (CSTI); CREST (Japan Science and Technology Agency) JPMJCR1671; JSPS KAKENHI Grant Number JP18K13469.

## Author contributions

K.M., T.S., and M.K. contributed to the initial conception of the ideas, to the working out of details, and to the writing and editing of the manuscript.

## Additional information

**Competing interests:** The authors declare no competing interests.

