## [Peer Review File · Nature Communications]

REVIEWERS' COMMENTS:

Reviewer #1 (Remarks to the Author):

This manuscript proves security of a variant of Phase-Matching QKD protocol, a family of the recently-proposed Twin-Field QKD protocol. These protocols arise considerable research efforts, because they may break the limits in the transmission distance. It has been long desired to complete the security proof considering the finite-length effect. The authors present a concise proof by introducing an approximation of the state satisfying operator dominance condition, which would be a powerful technique in this field. Because of the timeliness and effectiveness, I believe this manuscript is worth publishing in the Nature Communications.

The followings are my suggestions to improve the readability of the article:

1. The probability p_{even} is determined by the states Alice and Bob prepare. A simple explanation should be given why K_0^{even} coincides with the estimated value after Charlie's arbitrary measurement.
2. More detailed explanation on Figure 2, related to the manuscript would help understanding calculation procedure.
3. "real-linear combination" in p.3 is a little puzzling. It would be better to present explicitly as $\alpha^{(l)} \in \mathbb{R}$.
4. It would be better to give a definition of phase-randomized coherent state $|\tau(\mu)\rangle$ appeared in p. 4.

Reviewer #2 (Remarks to the Author):

Twin-Field QKD is a recently proposed protocol for quantum key distribution that is much more resistant to losses in the channel, in the sense that the key rate scales as the square root of the channel loss rather than a linear dependence. This allows us to do QKD over much larger distances using current technology. However, early proposals did not have full security proofs, and the current proofs either only yield key rates in the asymptotic regime and therefore cannot be used directly in practical scenarios, or give very poor bounds. This paper gives a new variant of the protocol with a proof of security in the finite block size regime which achieves reasonable key rates. In particular, for distances above ~150km, the key rates are substantially higher than even the asymptotic rates of decoy-state BB84.

This is a nice result that makes TFQKD directly applicable to experiments. However, optimizing the key rate is perhaps a bit incremental for a journal such as Nature Communications.

Reviewer #3 (Remarks to the Author):

In the submitted manuscript, entitled "Repeaterless quantum key distribution with efficient finite-key analysis overcoming the rate-distance limit", the authors modify the original PM-QKD protocol and propose a simple way to estimate the privacy amplification term in the finite-size regime. With the new way to estimate the phase error, the authors demonstrate the feasibility of modified PM-QKD protocol to break the PLOB bound in the finite key regime.

Recently, the single-photon-interference-based MDI-QKD is a heated topic, which owns the potential to break the secret-key-capacity bound and extend the QKD communication distance remarkably. Based on the thought to encode signals directly into the optical modes and apply former decoy-state techniques indirectly, the protocol named PM-QKD (Ref. [14]) is firstly proved to surpass the key rate bound. However, to apply the original decoy-state method, a sifting factor of 1/8 is sacrificed. To solve this problem, the improved PM-QKD analysis without phase randomization in the signal mode is provided (Ref. [17]). Nevertheless, to estimate the information leakage, one has to introduce different states other than the signal states, making the finite-size analysis hard.

The operator dominant condition combined with random sampling arguments, as proposed in the manuscript, is interesting, and can be seen as a good extension of decoy-state method, which provides new viewpoint for the parameter estimation in QKD. The work is meaningful and timely. I recommend its publication in Nature Communication. Below are some minor comments:

1. Note that the authors' names of Ref. [17] are "Lin and L\{u}tkenhaus".
2. In the mode "10"/"11", the rounds where Alice and Bob send different states, that is, Alice sends out vacuum while Bob sends out μ_1 light, are abandoned. Is it possible to utilize such signals to make the analysis tighter?
3. Is it possible to reduce the vacuum+3 intensities protocol in the manuscript, to a protocol with only two non-zero intensities, without violating the operator dominant requirement?

Point-to-point response to Reviewers #1 and #3.

Reviewer #1

1. *The probability p_{even} is determined by the states Alice and Bob prepare. A simple explanation should be given why K_0^{even} coincides with the estimated value after Charlie's arbitrary measurement.*

We added an intuitive explanation why the estimation is valid for any adversary on page 4.

2. *More detailed explanation on Figure 2, related to the manuscript would help understanding calculation procedure.*

We considerably expanded the caption of Fig. 2.

3. *"real-linear combination" in p.3 is a little puzzling. It would be better to present explicitly as $\alpha^{(i)} \in \mathbb{R}$.*

We corrected as the referee suggested.

4. *It would be better to give a definition of phase-randomized coherent state $\tau(\mu)$ appeared in p. 4.*

We add a definition as new Eq. (8) on page 4.

Reviewer #3

1. *Note that the authors' names of Ref. [17] are "Lin and Lütkenhaus".*

We corrected the errors.

2. *In the mode "10"/"11", the rounds where Alice and Bob send different states, that is, Alice sends out vacuum while Bob sends out μ_1 light, are abandoned. Is it possible to utilize such signals to make the analysis tighter?*

It is possible that the inclusion may improve the key rate, while currently we have no concrete evidence that it does improve. Since it requires detailed numerical study to derive a definite conclusion, we would like to leave it to future study. We added a comment on the Discussion section.

3. *Is it possible to reduce the vacuum+3 intensities protocol in the manuscript, to a protocol with only two non-zero intensities, without violating the operator dominant requirement?*

It is probable that the key rate stays positive when we make μ_2 closer to μ . On the other

hand, how much the key rate will decrease from that in the manuscript is currently unknown. We would like to leave the detail also to future study. We added a comment on the Discussion section.